# Recovering *Escherichia coli* Plasmids in the Absence of Long-Read Sequencing Data

**DOI:** 10.3390/microorganisms9081613

**Published:** 2021-07-28

**Authors:** Julian A. Paganini, Nienke L. Plantinga, Sergio Arredondo-Alonso, Rob J. L. Willems, Anita C. Schürch

**Affiliations:** 1Department of Medical Microbiology, University Medical Center Utrecht, 3584 CX Utrecht, The Netherlands; j.a.paganini@umcutrecht.nl (J.A.P.); N.L.Plantinga@umcutrecht.nl (N.L.P.); rwillems@umcutrecht.nl (R.J.L.W.); 2Department of Biostatistics, Faculty of Medicine, University of Oslo, 0372 Oslo, Norway; s.a.alonso@medisin.uio.no; 3Parasites and Microbes, Wellcome Sanger Institute, Cambridge CB10 1SA, UK

**Keywords:** WGS, plasmids, antibiotic resistance, bioinformatics, *Escherichia coli*

## Abstract

The incidence of infections caused by multidrug-resistant *E. coli* strains has risen in the past years. Antibiotic resistance in *E. coli* is often mediated by acquisition and maintenance of plasmids. The study of *E. coli* plasmid epidemiology and genomics often requires long-read sequencing information, but recently a number of tools that allow plasmid prediction from short-read data have been developed. Here, we reviewed 25 available plasmid prediction tools and categorized them into binary plasmid/chromosome classification tools and plasmid reconstruction tools. We benchmarked six tools (MOB-suite, plasmidSPAdes, gplas, FishingForPlasmids, HyAsP and SCAPP) that aim to reliably reconstruct distinct plasmids, with a special focus on plasmids carrying antibiotic resistance genes (ARGs) such as extended-spectrum beta-lactamase genes. We found that two thirds (*n* = 425, 66.3%) of all plasmids were correctly reconstructed by at least one of the six tools, with a range of 92 (14.58%) to 317 (50.23%) correctly predicted plasmids. However, the majority of plasmids that carried antibiotic resistance genes (*n* = 85, 57.8%) could not be completely recovered as distinct plasmids by any of the tools. MOB-suite was the only tool that was able to correctly reconstruct the majority of plasmids (*n* = 317, 50.23%), and performed best at reconstructing large plasmids (*n* = 166, 46.37%) and ARG-plasmids (*n* = 41, 27.9%), but predictions frequently contained chromosome contamination (40%). In contrast, plasmidSPAdes reconstructed the highest fraction of plasmids smaller than 18 kbp (*n* = 168, 61.54%). Large ARG-plasmids, however, were frequently merged with sequences derived from distinct replicons. Available bioinformatic tools can provide valuable insight into *E. coli* plasmids, but also have important limitations. This work will serve as a guideline for selecting the most appropriate plasmid reconstruction tool for studies focusing on *E. coli* plasmids in the absence of long-read sequencing data.

## 1. Introduction

*Escherichia coli* is a versatile micro-organism able to survive and thrive in different ecological habitats. It is a Gram-negative facultative anaerobe that commonly resides in the human gut as a commensal bacteria [1]. However, several members of this species also harbor the potential to cause severe infections, both intestinally [2] and extra-intestinally [3], in the healthcare settings [4] as well as in the community [5]. The ‘success’ of *E. coli* as a pathogen can be mostly attributed to the wide repertoire of virulence factors that strains may carry [6] and the increasing fraction of infections caused by multidrug-resistant strains [7]. Many of the antibiotic resistance genes and virulence factors present in *E. coli* are commonly encoded on plasmids, mobile genetic elements (MGE) that can be horizontally disseminated [8,9,10]. Therefore, precise identification and characterization of *E. coli* plasmids are highly relevant from an epidemiological and clinical standpoint.

Over the past decade, Illumina short-read sequencing platforms have become a popular technology to elucidate the genomic content and molecular epidemiology of bacteria. However, the frequent occurrence of repeat elements prohibits the assembly of complete replicons (plasmids and chromosomes) and often results in hundreds of contigs per genome with an unclear origin. Plasmid and chromosome contigs are mingled in draft genome assemblies, which challenges the accurate reconstruction of plasmids. More recently, long-read sequencing platforms (Oxford Nanopore and PacBio) have successfully resolved this issue, but short-read sequencing remains the *de facto* standard in many microbiology laboratories [11,12,13,14].

Several fully automated bioinformatics tools are currently available to predict bacterial plasmids from short-read sequencing data. Since 2018, at least 15 different tools have been created for this purpose (Appendix A). They can be broadly categorized into two main classes. The first class comprises software that produces a binary classification of contigs as either plasmid- or chromosome-derived, generating an output that predicts the complete plasmid content of a bacterial strain, often referred to as the ‘plasmidome’. An accurate plasmidome prediction has proven helpful to discover the genomic location of clinically relevant genes [15,16,17,18] and their role in shaping niche specificity [19], among others. The second class consists of tools that aim to recover distinct closed plasmid sequences. The output of these tools provides, in theory, a more comprehensive picture of the plasmid content of bacteria and allow to study the dissemination and epidemiology of specific plasmids [20].

Here, we reviewed the different tools and strategies to achieve binary prediction, for example fast k-mer based searches against reference plasmid databases (PlaScope and PlasmidSeeker), exploitation of the natural distribution bias of protein-coding genes between plasmids and chromosomes (Platon), and machine learning algorithms with different underlying features (cBAR, PlasFlow, mlplasmids, PlasClass, RFPlasmid and PPR-Meta) and others. Furthermore, we benchmarked six tools aimed at reconstructing fully closed distinct plasmids for use with *E. coli*, by using complete *E. coli* genomes that were recently deposited to public databases. The strategies applied by the reconstruction tools consist of graph-based approaches (plasmidSPAdes, gplas), reference-based approaches (MOB-Suite, FishingForPlasmids) and hybrid approaches which use reference- and graph information (HyAsP and SCAPP). We assessed their performance based on their ability to correctly recover different plasmids as distinct and complete predictions, including plasmids that carry clinically relevant antibiotic resistance determinants, such as extended-spectrum beta-lactamase (ESBL) genes.

## 2. Materials and Methods

### 2.1. Review of Plasmid Prediction Tools

We performed a systematic search of peer-reviewed publications deposited in PubMed by August 25th 2020, using the following search terms:

((plasmid*[Title])) AND ((software[Title/Abstract]) OR (tool*[Title/Abstract]) OR program[Title/Abstract])) AND ((predict*[Title/Abstract]) OR (sequencing[Title/Abstract]) OR (identification[Title/Abstract]) OR (prediction[Title/Abstract]) OR (contigs[Title/Abstract]) OR (assembly [Title/Abstract]) OR (NGS[Title/Abstract])).

This search resulted in 238 peer-reviewed publications that we manually curated to obtain a list of 17 different tools with the goal to study the plasmid content of bacteria in silico (Appendix A).

In order to find tools deposited on GitHub and GitLab, we used the search term ‘*plasmid*’. This resulted in 229 repositories from which 7 relevant tools were added to the selection (Appendix A). The Github location of FishingForPlasmids was obtained through personal communication with the developer.

### 2.2. Retrieving E. coli Complete Genomes and Metadata from NCBI Database

Ncbi-genome-download v0.2.10 (https://github.com/kblin/ncbi-genome-download/) was used to download all *E. coli* sequence labeled as ‘complete genomes’ up to 25 August 2020 (*n* = 1755). Metadata of the isolates was retrieved and parsed using Entrez-utilities v13.9 [21]. All scripts used to carry out the analyses in this study are available in a Git repository (https://gitlab.com/jpaganini/recovering_ecoli_plasmids).

### 2.3. Phylogenetic Analysis

Phylogroups were determined in silico by using ClermonTyping v1.4.0 [22]. Core- and accessory-genome distances were calculated by using PopPUNK v1.2 [23] with standard parameters. PopPUNK was also used to build a core-genome neighbor-joining tree with 1381 complete *E. coli* genomes downloaded from the NCBI database on 25 August 2020. Tree visualization and metadata information were integrated in Microreact [24] (Appendix A).

### 2.4. Benchmark Data Set Selection

Isolates that were not sequenced by both long- and short-read technologies (*n* = 559) were excluded, as well as sequences that were predicted as *Escherichia* cryptic clades [25] by in silico ClermonTyping (*n* = 12) and genomes that exhibited a predicted accessory-genome distance larger than 0.5 by PopPUNK (*n* = 2). We used a script written in R (version = 3.6.1) to remove genomes that had been used for developing the tested tools (*n* = 601). Moreover, we excluded genomes that did not carry any plasmids (*n* = 170), except for 19 randomly selected *E. coli* isolates without plasmids that were included as negative controls. In order to get a balanced data set, we removed a random sample of genomes isolated from farm animals (*n* = 161). Finally, we removed 30 genomes containing short-read-only assembled contigs that did not align to any replicon in their respective closed reference genome. The data set resulted in 240 *E. coli* complete genomes, which carried a total of 631 plasmids (Appendix A).

### 2.5. Evaluating Plasmid Diversity in Benchmarking Data

We used Mash v2.2.2 (k = 21, s = 1000) to estimate the pairwise k-mer distances of all plasmids (*n* = 3264) from all complete *E. coli* genomes (*n* = 1381). The obtained distances were clustered using the t-distributed stochastic neighbor embedding (t-SNE) algorithm with a perplexity value of 30, and data points (which represents individual plasmid sequences) were colored in orange if they were part of the benchmarking data set.

### 2.6. Plasmid Predictions

Illumina raw reads were downloaded using SRA Tools (v2.10.9). Reads were trimmed using trim-galore (v0.6.6) (https://github.com/FelixKrueger/TrimGalore to remove adapter contamination and bases with a phred quality score below 20. SPAdes (v3.14.0) [26] was applied to perform de novo assembly in careful mode and using kmer lengths of 37, 57 and 77. For isolates GCA_014117345.1_ASM1411734v1, GCA_006352265.1_ASM635226v1 and GCA_003812945.1_ASM381294v1, SPAdes was run using the --isolate option. The resulting contigs, assembly graphs and trimmed-reads were used as input for the different plasmid reconstruction tools, following the input requirements of the respective tools (Appendix A). All tools were run with default parameters. Tool’s versions were: FishingForPlasmids (no version information), MOB-suite (v3.0.0), SCAPP (v0.1.3), plasmidSPAdes (v3.14.0), gplas (v0.6.1), HyAsP (v1.0.0).

### 2.7. Analysis of the Plasmid Bins Composition

We used QUAST (v5.0.2) to align the contigs of each bin to the respective closed reference genome. An extended description of the parameters used is available at Appendix A. Based on the alignment results, we calculated precision, recall and F1-score as specified below.
$$Precision\;\left(\mathrm{bp}\right)=\frac{Alignment.length.against.reference.plasmid\;\left(\mathrm{bp}\right)}{Total.length.of.predicted.bin\;\left(\mathrm{bp}\right)}\;Recall\;\left(\mathrm{bp}\right)=\frac{Alignment.length.against.reference.plasmid\;\left(\mathrm{bp}\right)}{Total.length.of.reference.plasmid\;\left(\mathrm{bp}\right)}\;F1Score\;\left(\mathrm{bp}\right)=\frac{2\times Precision\;\left(\mathrm{bp}\right)\times Recall\;\left(\mathrm{bp}\right)}{Precision\;\left(\mathrm{bp}\right)+Recall\;\left(\mathrm{bp}\right)}$$

If a bin was composed of contigs derived from different plasmids, precision, recall and F1-score were reported for each plasmid-bin combination.

In order to quantify the chromosomal sequence content (if any) on a bin, we defined a chromosome contamination metric as follows.
$$Chromosomecontamination=\frac{Alignment.length.against.chromosome\;\left(\mathrm{bp}\right)}{Total.length.of.predicted.bin\;\left(\mathrm{bp}\right)}$$

### 2.8. Evaluating Maximum Theoretical Recall for Each Reference Plasmid

Depending on the input requirement of the respective tools (graph or contigs), we converted assembly graph nodes to FASTA format using the tool Any2Fasta (https://github.com/tseemann/any2fasta). or used the contigs produced by SPAdes and aligned them to their respective closed reference genomes using QUAST. Based on these alignments we calculated the maximum recall that could be obtained for reconstruction of every reference plasmid using short-read sequencing data (Appendix A).

### 2.9. Antibiotic Resistance Gene (ARG) Prediction

Resistance genes were predicted by running Abricate (v1.0.1) against the resfinder database (database indexed on 19 April 2020) with reference plasmids as query, using 80% as identity and coverage cut-off. The same software and parameters were used to predict the presence of ARGs in the plasmid bins generated by each of the plasmid reconstruction tools.

### 2.10. Evaluating Reconstruction of ARG Plasmids

For bins that carried ARGs, we calculated Recall_ARG_, as indicated below.
$$Recall\left(ARG\right)=\frac{Nr.of.correctly.predicted.ARGs.on.bin}{Total.nr.of.ARGs.on.reference.plasmid}$$

Bins that included the complete ARG content of the reference plasmid (Recall_ARG_ = 1) and were linked to the correct plasmid backbone (F1-score ≥ 0.95) were considered as correct reconstructions of the ARG-plasmid.

## 3. Results

### 3.1. Computational Methods to Predict the Plasmidome or Distinct Plasmids

We used a systematic search of peer-reviewed publications and two popular software-repository hosting web services and retrieved a total of 25 plasmid- or plasmidome- prediction tools (Appendix A). Most of the tools (*n* = 24) were fully automated and harbored the potential to be included in computational pipelines. Of these 24 tools, 13 tools were designed to analyze the plasmidome of multiple species using whole-genome sequencing data as input, while 8 tools can be applied to metagenomic sequences. A total of two tools, Recycler and RFPlasmid, worked with both types of input. Notably, we found one tool (FishingForPlasmids) that was developed to exclusively study the plasmid content of *E. coli*.

Based on the output, most of the tools (*n* = 23) can be broadly categorized into one of the following three classes. The first class comprises software that predicts the plasmidome, thus producing a binary classification of contigs as either plasmid- or chromosome-derived (*n* = 10). The second class consists of tools that aim to recover distinct plasmid sequences (*n* = 11) (Figure 1, Appendix A). The third class of tools seeks to facilitate the detection of known plasmids (*n* = 2). Below, we briefly review the computational strategies applied by 17 tools that belong to the first two categories. Four tools were excluded from this review for distinct reasons: plasmIDent uses long-reads as input, plasmidID and plasmidAssembler use a similar approach to MOB-suite for plasmid reconstruction and PLACNET requires manual intervention from the user.

#### 3.1.1. Binary Classification Tools

Binary classification tools take previously assembled contigs as input and classify them as being plasmid- or chromosome-derived.

PlaScope [27] and PlasmidPicker perform k-mer searches against reference plasmid databases. This strategy is very fast but limited to detecting k-mers that are present in the underlying database. Consequently, this produced high specificity and precision values but lower recall in a study that included a benchmark of PlaScope [27,28].

cBAR, PlasFlow and PlasClass all share a common underlying principle: using short k-mer frequencies and machine learning (ML) algorithms to classify metagenomic assemblies. More specifically, cBAR relies on observed differences in pentamer frequencies and uses a sequential minimal optimization (SMO) model. PlasFlow calculates the frequencies of multiple k-mers sizes (between 5 and 7 nt) and utilizes a neural-network voting classifier to integrate predictions. PlasFlow has a better performance than cBAR [29,30], but shows less reliable results for short contigs [31]. PlasClass addresses this issue by using a set of four logistic regression classifiers, each trained on sequences of different length [31]. Similar to cBAR, mlplasmids also relies on pentamer frequencies but uses a Support Vector Machine (SVM) model to determine the origin of contigs for a single species, and contains models for *Escherichia coli*, *Klebsiella pneumoniae* and *Enterococcus faecium*. Mlplasmids outperformed both cBAR and PlasFlow when classifying data derived from whole-genome sequencing experiments, and it can also accurately predict the plasmid localization of several antimicrobial resistance genes [29]. RFPlasmid [32], a recently released tool, uses a random forest classifier trained with a hybrid approach by identifying chromosomal and plasmids marker genes using two databases and also pentamer frequencies. This tool also works with metagenomic assemblies, albeit only for contigs from the 17 different species for which classifiers were trained. Platon exploits the natural distribution bias of protein-coding genes between plasmids and chromosomes and also analyzes higher-level characteristics of the contigs: circularization, presence of replication and mobilization proteins, presence of oriT and incompatibility sequences [28].

Finally, PPR-Meta [33] allows simultaneous identification of both phages and plasmids fragments from metagenomes by using a Convolutional Neural Network. Notably, instead of k-mer frequencies, this tool uses one-hot matrices to represent nucleotides and amino-acids sequences [33].

Despite the differences in approaches and performances, none of the aforementioned tools attempted to further sort the predicted plasmidome into individual plasmids. As a consequence, these tools are not suitable for studying the epidemiology of specific plasmids.

#### 3.1.2. Plasmid Reconstruction Tools

Based on their computational strategies, we can roughly subdivide plasmid reconstruction tools into three different categories: (i) de novo reconstruction of plasmids using assembly graph information, (ii) reference-based approaches and (iii) hybrid approaches.

PlasmidSPAdes, Recycler, metaplasmidSPAdes and gplas [34,35,36] perform a de novo reconstruction of plasmids using assembly graph information. PlasmidSPAdes and Recycler were released in 2016 and were the first tools that exploited the information on the assembly graph for identifying individual plasmids. PlasmidSPAdes is based on the assumption that plasmids have a different copy number than the chromosome, and therefore plasmid contigs will exhibit a different read coverage than chromosomal contigs. A number of studies have shown that this tool is able to reconstruct bacterial plasmids with high recall [11,37,38], but they have also revealed two major disadvantages of this approach: (1) plasmidSPAdes fails to identify large plasmids that have the same copy number as the chromosome and (2) it has a tendency to merge different plasmids together. Recycler also tries to identify plasmid-paths in the assembly graph by using coverage information but incorporates additional data regarding the topology of the selected paths. The main rationale behind this algorithm is that selected plasmid-paths should be cyclic, coverage should be homogeneous amongst all contigs and mated pair-end reads should map to the same path. Recycler appears to successfully identify short plasmids but yields very low precision values for long plasmids [11,37]. This issue is partially addressed by metaplasmidSPAdes, released in 2019 as an improvement on the original prediction algorithm of plasmidSPAdes. This tool allows prediction of dominant plasmids in metagenomes, defined as plasmids with coverage exceeding that of chromosomes and other plasmids. The algorithm iteratively extracts cyclic subgraphs with increasing coverage from the metagenome assembly graph. These potential plasmid sequences are later analyzed by a naive Bayesian classifier, called plasmidVerify, that further assesses the gene content of potential plasmids. None of the aforementioned tools takes advantage of the information embedded in the nucleotide sequences of the assembled contigs to *a priori* simplify the task of identifying plasmid subgraphs. In contrast, gplas initially classifies assembled contigs as plasmid-derived or chromosome-derived by using mlplasmids (or plasflow), a tool that exploits short k-mer frequencies for achieving such classification. Subsequently, plasmid-derived unitigs act as seeds for finding plasmid-walks with homogeneous coverage in the assembly graph, using a greedy approach. Gplas generates a plasmidome network in which nodes corresponding to plasmid unitigs and edges are created and weighted based on the co-existence of the nodes in the solution space of the computed walks. Finally, this plasmidome network is queried by a selection of network partitioning algorithms for generating bins of contigs that belong to the same plasmid [36].

MOB-suite and FishingForPlasmids use a reference-based approach for reconstructing individual plasmids. MOB-suite works as a modular set of tools for clustering, reconstruction and typing of plasmids from assemblies. This software initially uses Mash [39] and a single-linkage clustering algorithm to create clusters of similar plasmids present in a reference database. Input contigs are then aligned against this database using Blast and assigned to a plasmid cluster according to the best hits obtained. Contigs assigned to the same reference cluster constitute potential individual plasmid units. Also, the topology of the contigs is evaluated and every circular contig is considered an individual plasmid. Finally, each identified plasmid is queried against a different database for finding known replication and mobilization proteins and oriT sequences. According to the authors, MOB-suite performs better than plasmidSPades at correctly reconstructing plasmids from a benchmarking data set that included more than 370 plasmids from 14 different bacterial species [38]. However, the authors identified that MOB-suite splits single plasmids into different predictions more often than plasmidSPAdes. FishingForPlasmids attempts to reconstruct individual plasmids from *Escherichia coli* assemblies. This tool identifies plasmid-contigs by using BlastN to align each contig against a curated *E. coli* database. Each plasmid-derived sequence is further classified into discrete components by using a combination of plasmidFinder and pMLST [14].

Finally, HyAsP and SCAPP use a hybrid approach, mixing principles from reference-based and de novo methods. In HyAsP, a set of potential plasmid contigs is first selected based on: (1) a high density of known plasmid genes, identified by using a database, (2) high read coverage and (3) a length that does not exceed a maximum threshold. These plasmid-contigs will be used as seeds for finding plasmid-walks within the original assembly graph using a greedy algorithm. Plasmid-walks must satisfy the following conditions: (1) have a uniform GC content and sufficient read coverage, (2) do not have large gene-free segments and (3) total length of the plasmid-walk does not exceed a threshold. SCAPP, on the other hand, is designed for finding plasmids in metagenome assemblies. This algorithm starts by finding potential plasmid-contigs based on two strategies: (1) searching for plasmid-specific genes by using a curated database and (2) assigning weight to each contig based on the output from PlasClass, a ML-based binary classifier. The assembly graph is then queried to find cyclic walks of uniform coverage, similar to Recycler, but prioritizing the inclusion of contigs with strong evidence of plasmid-origin [40].

### 3.2. The Benchmark Data Set Represents the Diversity of Sequenced Plasmids

To benchmark the aforementioned plasmid reconstruction tools, we used a data set of 240 *E. coli* strains with complete genome sequences and short read data available from public databases that harbored 631 plasmids. These *E. coli* genomes were absent from all training data sets used to develop the selected plasmid prediction tools. The majority of the genomes derived from Europe (*n* = 170), Asia (*n* = 39) and North America (*n* = 24) (Figure 2A). They were isolated from multiple sources such as animals (*n* = 103), humans-clinical samples (*n* = 27), humans-community samples (*n* = 4), environmental sources (*n* = 86) and unknown sources (*n* = 13) (Figure 2B).

To assess if the selected genomes were a representative sample of the phylogenetic diversity of *E. coli*, we built a neighbor-joining tree combining our data set with 1141 complete *E. coli* genomes and determined the phylogroup of each of these genomes in silico. This analysis revealed that the selected genomes were distributed across the core-genome tree and that all phylogroups were represented with at least five strains. (Figure 2C).

Most of the genomes carried one (*n* = 73), two (*n* = 49) or three (*n* = 28) plasmids, but notably some genomes contained as much as nine (*n* = 3), ten (*n* = 1) or eleven (*n* = 1), with a median of two (mean = 2.62 plasmids). We found a clear bimodal plasmid size distribution, with peaks around 4500 bp and 100,000 bp (Figure 2D). Consequently, plasmids with a length smaller than 18,000 bp were classified as ‘small’ (*n* = 273), while plasmids that exceeded this cut-off value were classified as ‘large’ (*n* = 358).

Next, we wanted to assess the diversity of plasmids included in the benchmark data set. We used Mash to estimate the pairwise k-mer distances of all plasmids (*n* = 3264) from all complete *E. coli* genomes (*n* = 1381) and clustered them with the t-SNE algorithm. Plasmids included in this study were distributed among all major clusters, suggesting that this data set is able to properly capture the diversity of the *E. coli* pan-plasmidome currently available at NCBI (Figure 2E).

### 3.3. A Third of All Plasmids Could Not Be Correctly Reconstructed by Any of the Tools

We selected six tools to reconstruct distinct plasmid sequences. These tools applied different computational strategies: graph-based (plasmidSPAdes, gplas), reference-based (MOB-Suite, FishingForPlasmids) and hybrid (HyAsP and SCAPP).

The rest of the plasmid reconstruction tools were not included in the analysis because of a variety of reasons: Plasmid Assembler couldn’t be installed, plasmidID predictions were not completed due to errors during execution, PLACNET required manual intervention of the user, Recycler provided suboptimal results in comparison with plasmidSPAdes and HyAsP in previous studies [11,37] and metaplasmidSPAdes uses a similar approach to plasmidSPAdes but optimized for metagenomic samples.

We evaluated the predictions obtained with the six selected plasmid reconstruction tools in terms of (i) speed and memory requirements, (ii) the number of plasmid predictions, (iii) correct reconstruction of reference plasmids, (iv) chromosomal contamination included in predicted plasmids, and (v) correct reconstruction of ARG-plasmids.

We used a High-Performance Cluster (HPC) to run the tools with minimal resources (number of cores = 2, 4GB of RAM per genome), and documented the total CPU-time and memory required by each of them (Table 1, Appendix A). Most tools required less than 100 CPU hours to complete all predictions, except for plasmidSPAdes which used 321.07 CPU hours. In contrast, FishingForPlasmids was the fastest tool and completed the task in 10.60 CPU hours. PlasmidSPAdes and SCAPP had the highest memory requirements, utilizing a total of 442.03 Gb and 435.23 Gb of RAM, respectively. The remaining tools required less than 300 Gb to complete all predictions. Notably, FishingForPlasmids only required a total of 36.57 Gb.

Next, we evaluated the number of plasmid predictions produced by each tool and calculated the difference between this number and the true number of plasmids present in the benchmark data set (Table 1, Appendix A). The total number of plasmid predictions ranged from 377 (FishingForPlasmids) to 2590 (HyAsP). plasmidSPAdes, MOB-suite, SCAPP and HyAsP overestimated the true number of plasmids (*n* = 631), while gplas and FishingForPlasmids underestimated this number. PlasmidSPAdes displayed the least deviation by producing 642 bins, and therefore exceeding the total number of plasmids by 11. Nevertheless, these absolute numbers do not reflect whether predictions were correct or incorrect.

In order to evaluate how the different tools performed at recovering *E. coli* plasmids as distinct and complete predictions, we studied the distributions of recall, precision and F1-score (Table 1, Appendix A) for all plasmid predictions made by the tools. Based on these results, we determined an F1-score cut-off value of 0.95 to define a plasmid as correctly reconstructed (or recovered) (Appendix A).

MOB-suite correctly recovered 317 (50.24%) plasmids (F1-score ≥ 0.95), including 70 (11.10%) that couldn’t be reconstructed by any other software (Figure 3A,B, Table 1). Similarly, plasmidSPAdes reconstructed a total of 263 (41.68%) plasmids, including 55 (8.72%) that were not recovered by other tools. Interestingly, 14 of these ‘unique reconstructions’ were also missing from the short-read assembly graphs (Appendix A). The rest of the tools achieved smaller quantities of correct plasmid reconstructions, with values ranging from 92 (14.58%) to 152 (24.09%) (Figure 3A,B, Table 1).

We found that a total of 418 (66.25%) plasmids were correctly reconstructed by at least one of the tools (Figure 3C). Out of these, only 7 (1.11%) were reconstructed by all tools concurrently, 273 (43.26%) by multiple tools and 138 (21.9%) by a single tool. Interestingly, combining MOB-suite and plasmidSPAdes predictions together achieved the correct reconstruction of 400 (63.39%) plasmids, and incorporating the predictions from the remaining tools only resulted in the reconstruction of 18 (2.85%) additional plasmids. Notably, a total of 213 (33.75%) plasmids were incorrectly reconstructed (F1 score < 0.95) by all tools, including 21 (3.32%) that were not even detected. The majority of ARG-plasmids (*n* = 85, 57.8%) could not be correctly reconstructed by any of the tools (Appendix A).

We also compared the performance of the software when attempting to reconstruct small- and large plasmids separately. For small plasmids, we discovered that all tools displayed similar F1-score distributions, with medians ranging from 0.95 to 0.99. However, the tools did not detect 21.25–89.74% of small plasmids (Appendix A). PlasmidSPAdes and MOB-suite were the only tools that achieved the correct reconstruction of most of these replicons, with a total of 168 (61.54%) and 155 (55.31%), respectively (Table 1). When considering the reconstruction of large plasmids, percentages of not-detected plasmids were much lower and ranged from 2.23% to 20.11% across tools. MOB-suite exhibited the highest F1-score values (median = 0.74, IQR = 0.17–0.97) and correctly reconstructed 166 (46.3%) of these replicons, significantly surpassing the reconstruction capacity of the rest of the tools, which ranged from 45 (12.57%) to 95 (26.54%) (Table 1, Appendix A*)*. Not surprisingly, most tools correctly reconstructed a higher fraction of small plasmids, and also displayed higher F1-score values (Table 1, Appendix A) when comparing with the reconstruction of large plasmids. FishingForPlasmids was the only exception as it recovered a total of 14 (5.13%) small and 78 (21.79%) large plasmids.

All tools incorrectly incorporated chromosome-derived sequences into their predictions (Appendix A, Table 1). FishingForPlasmids performed best at avoiding this error, and only 7 (1.8%) predictions contained chromosomal contamination. In contrast, HyAsP introduced chromosomal contigs in 1340 (51.7%) predictions with a chromosome contamination median of 0.88 (IQR = 0.5–0.99), including 1251 pure chromosome bins (chromosome contamination = 1). Notably, plasmidSPAdes and MOB-suite had a similar proportion of contaminated bins, 295 (46%) and 297 (40.2%), yet with different chromosome contamination medians of 0.75 (IQR = 0.14–0.92) and 0.10 (IQR = 0.03–0.99), respectively. Out of these, MOB-suite produced 65 predicted bins which exclusively consisted of chromosome sequences, while plasmidSPAdes generated 20 of them. SCAPP introduced chromosomal sequences in 249 (25.2%) predictions, but notably only 1 of them was only composed of chromosome sequences. Finally, gplas incorporated chromosomal sequences in 197 (35.8%) predictions, of which 70 were exclusively composed of these types of sequences.

### 3.4. Plasmids Carrying Antibiotic Resistance Genes Were Difficult to Reconstruct for All Tools

Our data set included 147 (23.3%) plasmids containing antibiotic resistance genes (ARG-plasmids), carrying a total of 618 resistance genes. Most of these replicons carried one (*n* = 43), two (*n* = 17), three (*n* = 12) or four (*n* = 17) ARGs (Appendix A). Interestingly, plasmids carrying ARGs had a median length of 109,773 bp (IQR = 83,300–132,865 bp), and were markedly larger than plasmids with no resistance determinants (median length 6930 bp; IQR = 4072–91,111 bp). Furthermore, 143 (97.2%) ARG-plasmids were classified as large, while only 4 (2.8%) were small plasmids (Appendix A).

To investigate how the tools performed at reconstructing ARG-plasmids, we analyzed Recall, Precision and F1-score values for these replicons (Appendix A). Furthermore, we extracted the bins that contained antibiotic resistance genes, and explored the fraction of detected ARGs in each prediction -Recall(ARG)-. An ARG-plasmid was considered as correctly reconstructed if the prediction simultaneously included all ARGs -Recall(ARG) = 1-and correctly represented the reference plasmid backbone (F1-score ≥ 0.95).

We discovered that the reconstruction of large ARG-plasmids was particularly challenging for the evaluated tools, since all of them exhibited lower F1-score values in comparison with the reconstruction of large non-ARG-plasmids (Appendix A, Table 1). We excluded small plasmids from this comparison due to the low amount of small ARG-plasmids present in our data set.

MOB-suite correctly identified 548 (88.67%) plasmid-derived ARGs, and achieved 41 (27.89%) correct ARG-plasmid reconstructions (Figure 4A,B, Table 1). In 49 (33.3%) additional predictions, all ARGs were assigned into a single bin -Recall(ARG) = 1-, but the bin incorrectly represented the reference plasmid backbone (F1-score < 0.95) (Figure 4C) by being incomplete, hybridized with sequences derived from other replicons, or both (Appendix A). Moreover, we discovered that MOB-suite incorrectly incorporated 92 chromosome-derived ARGs, distributing them among 39 bins. Finally, we found that when predicting large ARG-plasmids, this tool presented remarkably lower recall values (median = 0.38, IQR = 0.09–0.88) in comparison with reconstruction of large non-ARG-plasmids (median = 0.87, IQR = 0.19–0.98) (Appendix A).

PlasmidSPAdes detected 390 (63.11%) plasmidderived ARGs, and correctly reconstructed 23 (15.65%) ARG-plasmids. Additionally, in 59 (40.14%) predictions all ARGs were assigned to a single bin, but the plasmid backbone was most frequently contaminated with sequences from other replicons (Appendix A). Notably, this tool couldn’t predict any of the ARGs present in 37 (25.17%) reference ARG-plasmids (Figure 4A,B, Table 1). Finally, for the reconstruction of large ARG-plasmids, plasmidSPAdes presented remarkably lower precision values (median = 0.47, IQR = 0.31–0.92) in comparison with reconstruction of large non-ARG-plasmids (median = 0.9, IQR = 0.35–1) (Appendix A).

The rest of the tools successfully reconstructed smaller fractions of ARG-plasmids, ranging from 5 (3.4%) to 13 (8.84%). Interestingly, HyAsP detected a high fraction of plasmid-derived ARGs (*n* = 525, *n* = 84.95%), but it only achieved the correct reconstruction of 5 (3.4%) ARG-plasmids. For most HyAsP predictions, all ARGs couldn’t be assigned to a single bin (*n* = 66, 44.9%) or presented an incorrect plasmid backbone (*n* = 62, 42.18%). FishingForPlasmids detected the least amount of resistance genes (*n* = 133, 21.52%) and couldn’t predict any of the ARGs present in 97 (66%) reference ARG-plasmids.

Next, we evaluated the performance of the tools when reconstructing plasmids that carry ESBL genes (ESBL plasmids). Our data set included 60 ESBL plasmids, each carrying a single ESBL gene. Most abundant ESBL variants were CTX-M15 (*n* = 16, 25%), CTX-M55 (*n* = 12, 20%) and CTX-M1 (*n* = 6, 10%) (Appendix A). Furthermore, we observed that ESBL genes were harbored by plasmids with diverse sequences (Appendix A).

MOB-suite correctly identified a total of 57 (95%) ESBL genes of plasmid origin, of which 20 were also assigned to the correct plasmid backbone (F1-score ≥ 0.95), resulting in a 33% correct reconstruction of the ESBL plasmids (Table 1, Appendix A). Despite this, MOB-suite predictions achieved high F1-scores for reconstruction of ESBL plasmids (median = 0.93, IQR = 0.72–0.97) (Table 1, Appendix A).

The rest of the tools reconstructed ESBL plasmids with less success, ranging from 0 (0%) to 10 (16.67%) total correct reconstructions (Table 1, Appendix A). HyAsP detected a high fraction of plasmid-derived ESBL genes (*n* = 52, 86.67%), but did not achieve the correct reconstruction of any plasmids. PlasmidSPAdes detected the majority of plasmid-derived ESBL genes (*n* = 40, 66.66%), and these were included in bins that presented high recall (median = 0.97, IQR = 0.77–0.96) but low precision values (median = 0.52, IQR = 0.38–0.95) (Table 1, Appendix A).

## 4. Discussion

A tool that is able to correctly predict *E. coli* plasmids will assist in identifying clinically relevant plasmids [41,42,43,44] and improve our understanding of the complex dynamics of ARG dissemination across different ecological niches [45,46,47]. From the vast offer of software to predict plasmids from short-read data we selected six tools and benchmarked their performances when attempting to reconstruct individual *E. coli* plasmids, with a special focus on plasmids that carry ARGs.

A total of 418 (66.24%) plasmids were correctly reconstructed by at least one of the tools compared in this benchmark. Interestingly, 400 (63.39%) of these plasmids were recovered by combining the predictions from MOB-suite and plasmidSPAdes alone. Therefore, adding the predictions from the rest of the tools resulted only in 18 (2.85%) additional correct reconstructions.

We observed that plasmidSPAdes correctly reconstructed the highest fraction of small plasmids (*n* = 168, 61.5%). This result is consistent with the observations that small plasmids usually have high copy numbers [48] and therefore exhibit a higher coverage; which in theory would facilitate their prediction using this tool. A similar success at predicting small plasmids was also reported by [11,38]. Nevertheless, it is worth noticing that most small plasmids (*n* = 215, 79%) are represented as a single node in the assembly graph. Therefore, using a binary classification tool would be sufficient for correctly predicting these replicons.

MOB-suite correctly reconstructed a total of 166 (46.37%) large plasmids, and considerably outperformed the rest of the tools, which ranged from 45 (12.57%) to 95 (26.54%) correct reconstructions. Nevertheless, MOB-suite’s performance strongly depends on its underlying database, which is enriched for *Enterobacteriaceae* plasmid sequences [38]. Consequently, the reconstruction capacity of this tool could be different when attempting to predict plasmids from bacterial species less frequently represented in its database.

A third (*n* = 213, 33.76%) of all plasmids could not be correctly reconstructed by any of the evaluated tools. In particular, the reconstruction of ARG-plasmids proved to be problematic. We hypothesize that ARG-plasmids constitute a particularly hard puzzle to solve for all compared computational approaches, for several reasons.

Firstly, ARG-plasmids usually carry a high number of repeated sequences [49,50,51,52], and therefore exhibit highly entangled assembly graphs. Secondly, ARGs are frequently located on large plasmids with low copy number, and therefore have coverage values that are similar to chromosomes [48,52]. Consequently, finding plasmid-walks with differential coverage in the assembly graphs could be challenging for all tools relying on this strategy. This hypothesis is supported by the observation that plasmidSPAdes predicted large ARG-plasmids with the lowest precision values (median = 0.47, IQR = 0.31–0.92) of all tools, indicating that these plasmids are more frequently merged with sequences derived from other replicons. Additionally, this tool failed to predict 37% of all plasmid-located ARGs, which would be explainable in case that these contigs should have coverage values similar to the chromosomes.

Thirdly, ARG-plasmids are frequently built as mosaic-like structures, containing mobile components that can be found in different plasmid backbones [48,52,53,54,55]. This type of genomic organization also complicates their reconstruction using reference-based methods, since databases might contain very similar fragments that are shared by a variety of plasmids. Consequently, unequivocally assigning these “shared fragments” to a unique reference plasmid (or plasmid group) could be problematic. This is supported by the results obtained using MOB-suite. This software identified the highest proportion of plasmid-derived ARGs (*n* = 548, 88.67%), but most ARG-plasmids reconstructions had either an incomplete ARG content (*n* = 47, 31.97%) or an incorrect backbone (*n* = 49, 33.33%). These results, in combination with the low recall values observed (median = 0.38, IQR = 0.09–0.88) seems to suggest that large ARG-plasmids were frequently split into multiple bins.

Despite the aforementioned limitations, MOB-suite was the most effective tool at predicting ARG-plasmids in *E. coli*, achieving the correct reconstruction of 41 (27.89%) of these, while the rest of the tools ranged from 5 (3.4%) to 23 (15.65%) correct ARG-plasmid reconstructions. Additionally, MOB-suite was the best performing tool for prediction of ESBL-plasmids. It identified 57 (95%) plasmid-borne ESBL-genes and had a median F1-score of 0.93 (IQR = 0.72–0.97). However, it must be noted that a fraction (*n* = 13, 22.80%) of ESBL-plasmid predictions presented low F1-score values, implying that in these cases the contigs carrying the ESBL gene were associated with the incorrect plasmid backbone.

All tools exhibited chromosomal contamination in their predictions. Notably, FishingForPlasmids outperformed the rest of the tools and only included chromosomal sequences in 7 (1.8%) bins. The rest of the tools included chromosomal sequences in a range from 25.25% to 51.73% of the bins. Surprisingly, MOB-suite included chromosomal sequences in 297 (40.2%) bins, including 65 chromosome-only predictions (chromosome contamination = 1).

A fraction of the plasmids (*n* = 28, 4.4%) were completely absent (recall = 0) from contig sequences and nodes in the assembly graph. Interestingly, 14 of these replicons were correctly reconstructed by plasmidSPAdes when using pair-end reads as input. This suggests that the quality of the assembly has impacted the ability of the tools to reconstruct certain plasmids. Consequently, it is possible that plasmid predictions for *E. coli* could be optimized by running SPAdes with different parameters, by performing assembly with different assemblers or through construction of Illumina libraries with a different read length.

The results from our study indicate that accurate reconstruction of *E. coli* plasmids from short-reads is still challenging using currently available bioinformatic methods. Long reads generated by Oxford Nanopore or PacBio technologies can span repeat elements in the bacterial genomes and are therefore useful to obtain complete plasmid sequences. However, long-reads still exhibit a lower sequencing accuracy than Illumina reads [56], and small plasmids (size < 10 kb) are frequently underrepresented or absent in Nanopore libraries [57,58]. Consequently, combining long- and short-read sequences is currently the best option for correctly reconstructing *E. coli* plasmids. Nevertheless, the accuracy of long-reads has been increasing in recent years, mainly due to the release of improved hardware and also owing to the development of bioinformatic tools designed for read error correction [56]. It is possible that in the near future long-read only assemblies will provide the best alternative for obtaining complete bacterial genomes.

Nonetheless, in the absence of long-reads, bioinformatic tools can be applied to gain valuable insight on different aspects of the plasmidome of *E. coli.* MOB-suite presented the best overall performance of all tools, but predictions were frequently contaminated with chromosomal sequences. Consequently, using MOB-suite coupled to a binary classification tool could improve plasmid predictions in *E. coli*. Furthermore, these predictions could be used as an initial screening step for selecting interesting isolates for long-read sequencing.

## Figures and Tables

**Figure 1 microorganisms-09-01613-f001:**
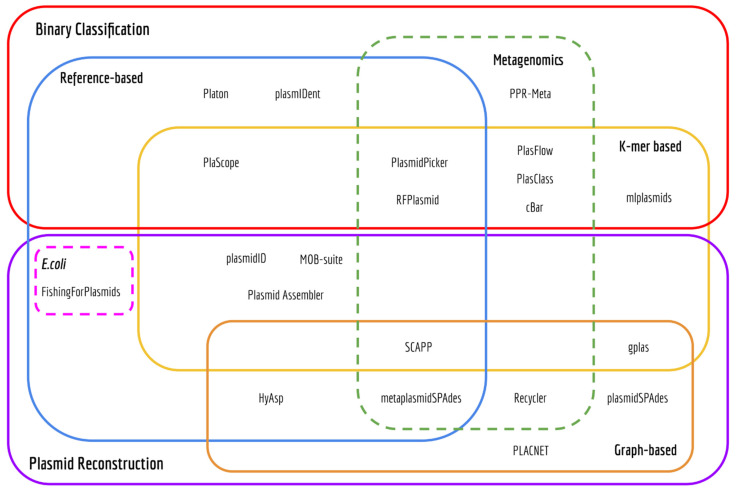
Euler diagram of bioinformatics tools to predict the plasmidome of bacteria.

**Figure 2 microorganisms-09-01613-f002:**
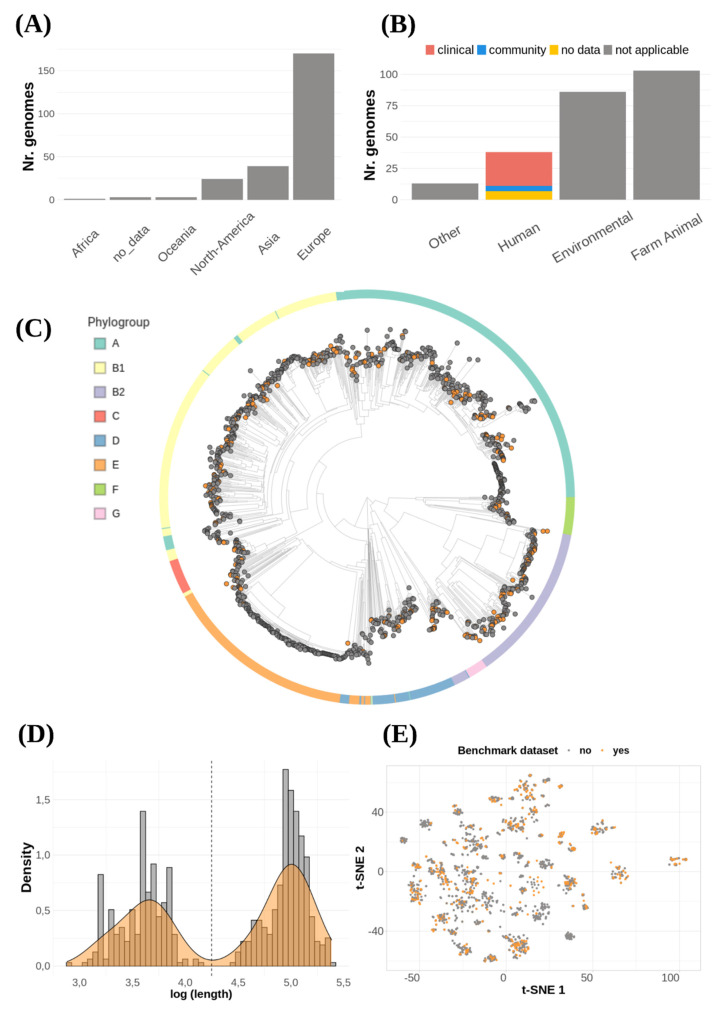
(**A**) Genomes distribution according to geographical location and (**B**) isolation source. (**C**) Core-genome clustering constructed using PopPUNK. We included 1381 complete *E. coli* genomes available at NCBI database. Orange tips (*n* = 240) indicate genomes that were included in the benchmarking data set, and outer colors indicate phylogroups. (**D**) Plasmid length histogram and density plot. Dashed line indicates the cut-off length (18,000 bp) for considering a plasmid as small or large (**E**) tSNE plot based on plasmids k-mer distances obtained with MASH (k = 21, s = 1000). Plasmids included in this benchmark (*n* = 631) are colored in orange.

**Figure 3 microorganisms-09-01613-f003:**
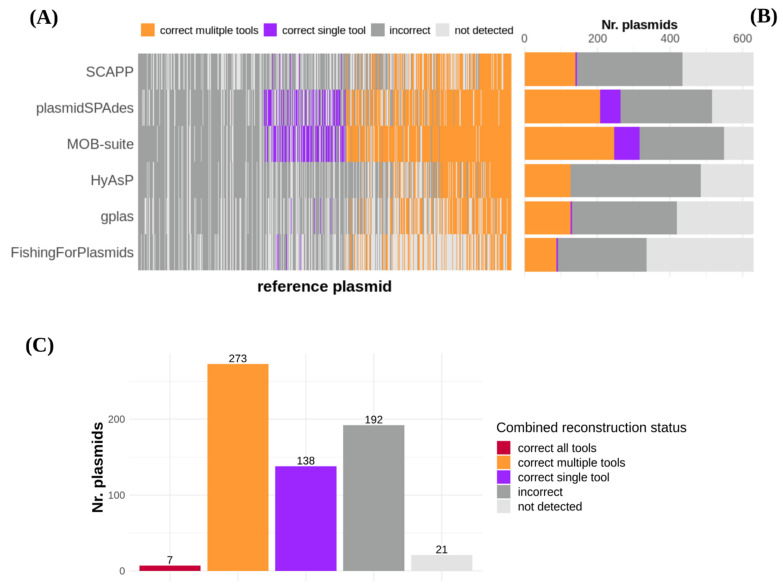
(**A**) Reconstruction performance of each tool for all reference plasmids. Reference plasmids have been ordered according to the number of tools by which they were correctly reconstructed, from low (left; reconstructed by 0/6 tools) to high (right; reconstructed by 6/6 tools). Plasmids that were reconstructed with an F1-score ≥ 0.95, were considered as correct reconstructions. Plasmids for which no contig was included in the predictions were considered as ‘not-detected’. (**B**) Absolute count of all reconstruction status achieved by each tool. (**C**) Absolute count of reconstruction categories when combining predictions from all tools.

**Figure 4 microorganisms-09-01613-f004:**
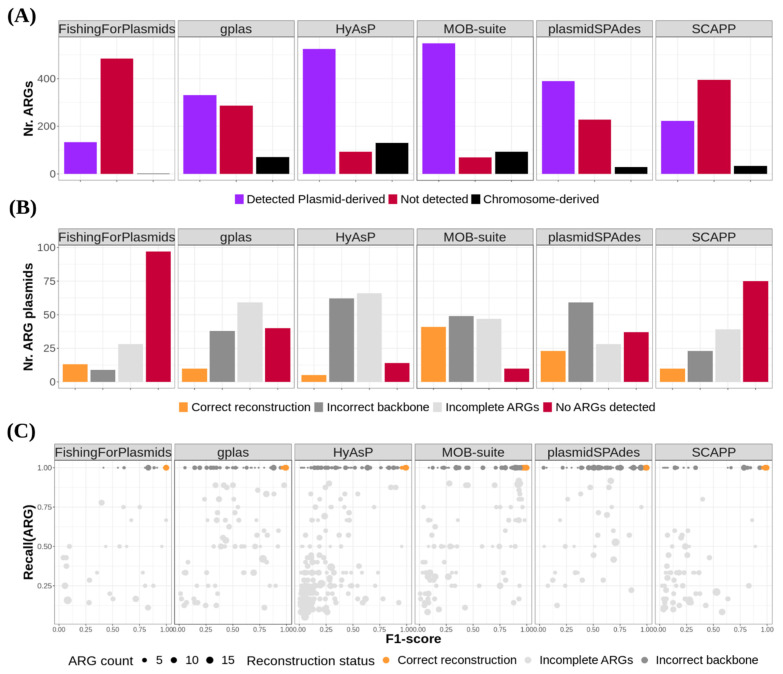
(**A**) Bar plot displaying the number of plasmid-derived ARGs that were detected/not detected by each of the tools. This plot also shows the number of chromosome derived ARGs included in the plasmid predictions. (**B**) Bar plot displaying the number of reference ARG-plasmids belonging to each different reconstruction category. Reconstruction categories were defined as follows. Correct reconstruction: all ARGs were predicted in the same bin (Recall(ARG) = 1) and the backbone of the plasmid was correct (F1-score ≥ 0.95). Incorrect backbone: all ARGs were predicted in the same bin (Recall(ARG) = 1) but the backbone of the plasmid was incorrect (F1-score < 0.95). Incomplete ARGs: Not all ARGs were included in the same bin (Recall(ARG) < 1). No ARGs detected: None of the ARGs derived from the reference plasmids were included in any bins created by the tool. (**C**) Scatter-plot showing relation between Recall(ARG) and F1-score (bp) values for predictions that carry at least one ARG of plasmid origin. Dots are colored according to the same criteria as in B.

**Table 1 microorganisms-09-01613-t001:** Summary of tool’s performances.

	HyAsP	MOB-Suite	gplas	plasmidSPAdes	SCAPP	FishingFor Plasmids
**Computational Performance**						
Memory Usage (GB)	299.2	202.82	150.36	442.03	435.23	36.57
CPU-Time (hr)	46.57	46.62	83.64	321.07	70.96	10.6
**Nr. of plasmid predictions**						
Nr. total predicted plasmids (bins)	2590	738	550	642	986	377
Nr. correct predictions of plasmid absence (%)	2 (10.53)	13 (68.42)	17 (89.47)	9 (47.37)	17 (89.47)	18 (94.74)
**Plasmids reconstruction** **All Plasmid (*n* = 631)**						
Nr. correctly reconstructed plasmids (%)	127 (20.13)	317 (50.24)	130 (20.6)	263 (41.68)	152 (24.09)	92 (14.58)
Nr. small plasmids (%)	82 (30.04)	151 (55.31)	87 (31.87)	168 (61.54)	98 (35.9)	14 (5.13)
Nr. large plasmids (%)	45 (12.57)	166 (46.37)	43 (12.01)	95 (26.54)	54 (15.08)	78 (21.79)
Nr. incorrectly reconstructed plasmids (%)	358 (56.74)	231 (36.61)	289 (45.8)	252 (39.94)	291 (46.12)	243 (38.51)
Nr. small plasmids (%)	53 (19.41)	50 (18.32)	17 (6.23)	47 (17.22)	59 (21.61)	14 (5.13)
Nr. large plasmids (%)	305 (85.20)	181 (50.56)	272 (75.98)	205 (57.26)	232 (64.80)	229 (63.97)
Nr. undetected plasmids (%)	146 (23.14)	83 (13.15)	212 (33.6)	116 (18.38)	188 (29.79)	296 (46.91)
Nr. small plasmids (%)	138 (50.55)	72 (26.37)	169 (61.9)	58 (21.25)	116 (42.49)	245 (89.74)
Nr. large plasmids (%)	8 (2.23)	11 (3.07)	43 (12.01)	58 (16.2)	72 (20.11)	51 (14.25)
F1-score (median-IQR) *	0.12 (0.04–0.41)	0.89 (0.3–0.98)	0.59 (0.3–0.94)	0.95 (0.49–0.99)	0.18 (0.07−0.81)	0.64 (0.29–0.93)
Small plasmids *	0.98 (0.76–0.99)	0.98 (0.94–0.99)	0.99 (0.98–0.99)	0.98 (0.96–0.99)	0.96 (0.88–0.99)	0.95 (0.7–0.98)
Large plasmids *	0.11 (0.04–0.32)	0.74 (0.17–0.97)	0.49 (0.21–0.76)	0.6 (0.31–0.97)	0.12 (0.06–0.41)	0.61 (0.28–0.91)
Recall (median-IQR) *	0.07 (0.02–0.32)	0.89 (0.21–0.99)	0.5 (0.22–0.93)	0.99 (0.88–1)	0.13 (0.04–0.78)	0.51 (0.18–0.93)
Small plasmids *	1 (0.92–1)	1 (0.96–1)	1 (0.98−1)	1 (1–1)	0.99 (0.92–1)	1 (0.96−1)
Large plasmids *	0.06 (0.02–0.2)	0.63 (0.12–0.96)	0.4 (0.16–0.72)	0.94 (0.36–0.99)	0.07 (0.03–0.31)	0.46 (0.16–0.84)
Precision (median-IQR) *	0.87 (0.5–0.98)	0.98 (0.68–1)	0.97 (0.55–1)	0.93 (0.41–0.98)	0.8 (0.39–0.94)	1 (1−1)
Small plasmids *	0.96 (0.86–0.98)	0.98 (0.95–0.99)	0.98 (0.97–0.99)	0.96 (0.92–0.98)	0.95 (0.83–0.98)	0.96 (0.65–0.97)
Large plasmids *	0.84 (0.48–0.98)	0.97 (0.53–1)	0.93 (0.47–1)	0.58 (0.33–0.99)	0.75 (0.34–0.92)	1 (1–1)
Chromosome contamination (Median-IQR)	0.88 (0.59–0.99)	0.1 (0.03–0.99)	0.45 (0.11–1)	0.75 (0.14–0.92)	0.3 (0.09–0.66)	1 (0.6–1)
Nr. bins with chromosome contamination (%)	1340 (51.73)	297 (40.2)	197 (35.81)	295 (45.95)	249 (25.25)	7 (1.86)
Nr. pure chromosome bins	1251	65	70	20	1	4
**Plasmids reconstruction** **ARG-plasmids (*n* = 147)**						
ARGs in bins						
Nr. plasmid-derived ARGs (%)	525 (84.95)	548 (88.67)	331 (53.56)	390 (63.11)	223 (36.08)	133 (21.52)
Nr. chromosome-derived ARGs	130	92	71	29	34	1
Reconstruction status						
Nr. plasmids correctly reconstructed (%)	5 (3.4)	41 (27.89)	10 (6.8)	23 (15.65)	10 (6.8)	13 (8.84)
Nr. plasmids predicted with incorrect backbones (%)	62 (42.18)	49 (33.33)	38 (25.85)	59 (40.14)	23 (15.65)	9 (6.12)
Nr. plasmids predicted with incomplete ARG content (%)	66 (44.9)	47 (31.97)	59 (40.14)	28 (19.05)	39 (26.53)	28 (19.05)
Nr. plasmids with no ARGs predicted (%)	14 (9.52)	10 (6.8)	40 (27.21)	37 (25.17)	75 (51.02)	97 (65.99)
Large ARG-plasmids reconstruction metrics (*n* = 143)						
Recall (Median-IQR) *	0.06 (0.02–0.16)	0.38 (0.09–0.88)	0.29 (0.14–0.62)	0.87 (0.2–0.96)	0.06 (0.03–0.17)	0.35 (0.15–0.55)
Precision (Median-IQR) *	0.84 (0.46–0.99)	0.92 (0.42–1)	0.86 (0.44–1)	0.47 (0.31–0.92)	0.71 (0.32–0.88)	1 (1–1)
F1-score (Median-IQR) *	0.1 (0.04–0.26)	0.44 (0.13–0.9)	0.41 (0.19–0.65)	0.53 (0.24–0.73)	0.1 (0.05–0.26)	0.51 (0.25–0.69)
Nr. detected plasmids (%)	141 (98.60)	141 (98.60)	135 (94.41)	129 (90.21)	113 (79.02)	138 (96.50)
**Plasmids reconstruction ESBL-plasmids (*n* = 60)**						
ESBL genes in bins						
Nr. plasmid-derived (%)	52 (86.67)	57 (95)	27 (45)	40 (66.67)	23 (38.33)	11 (18.33)
Nr. chromosome-derived (%)	10	8	7	2	2	0
Reconstruction status						
Nr. ESBL genes in correct plasmid backbone (%)	0 (0)	20 (33.33)	4 (6.67)	10 (16.67)	5 (8.33)	6 (10)
Nr. ESBL genes in incorrect plasmid backbone (%)	52 (86.67)	37 (61.67)	23 (38.33)	30 (50)	18 (30)	5 (8.33)
Reconstruction metrics						
F1-score (Median-IQR) *	0.29 (0.07–0.46)	0.93 (0.72–0.97)	0.69 (0.45–0.88)	0.65 (0.51–0.95)	0.27 (0.09–0.84)	0.98 (0.71–0.99)
Recall (Median-IQR) *	0.18 (0.04–0.31)	0.89 (0.77–0.96)	0.65 (0.36–0.84)	0.96 (0.89–0.97)	0.23 (0.05–0.84)	0.95 (0.56–0.99)
Precision (Median-IQR) *	0.91 (0.54–0.98)	0.98 (0.93–1)	0.97 (0.89–1)	0.52 (0.38–0.95)	0.85 (0.72–0.92)	0.99 (1–1)

* In all cases, undetected plasmids were not included in the calculation of Precision, Recall and F1-score.

## Data Availability

The complete code and files required to reproduce the analysis of this study are publicly available at GitLab under a GPL3.0 license (https://gitlab.com/jpaganini/recovering_ecoli_plasmids).

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
