# Peer review of "Recovering Escherichia coli Plasmids in the Absence of Long-Read Sequencing Data"

_microorganisms, 2021, doi:10.3390/microorganisms9081613_

Round 1

Reviewer 1 Report

The authors compared the performance of several tools in plasmids reconstruction for small reads sequencing programs. Obviously, the main problem is on length that impaired proper assembly and reconstruction. The work is well performed, my only concern is on abstract composition(too much descriptive and long).

Reviewer 2 Report

The work of Paganini et. al., describes a comprehensive review of plasmid reconstruction tools and a benchmarking analysis for 6 of them. The work is well planned, executed, described hence it was a pleasure to read it. I believe that the paper is ready for publication (with or without responding to the points below)

Points that could be included in the discussion:

1. All tools use Spades output. How does the quality of Spades assembly, which could be affected by the type of NGS data (eg. single end vs paired end reads, reads of different length) affect the plasmid prediction? Some of the plasmids (~5%) are not a part of contigs or nodes on graphs. In other words, is there a room for improvement at this stage?

2. What is the future of single-read only based plasmid reconstruction? (please speculate)

Minor remarks

- lines 155-183 please use underscores or dots to separate words in the equations, it would be easier to read them

- line 425 – please include the reference to Figure S6 earlier in the section
